# LieRE: Generalizing Rotary Position Encodings to Higher Dimensional Inputs

## Abstract

Rotary Position Embeddings (RoPE) have demonstrated efficacy and gained widespread adoption in natural language processing. However, their application to other modalities has been less prevalent. This study introduces Lie group Relative position Encodings (LieRE), which extend beyond RoPE by accommodating n-dimensional inputs. LieRE encodes positions of tokens by replacing the RoPE rotation matrix with a dense, high-dimensional, rotation matrix generated via a learned map. We conducted empirical evaluations of LieRE on 2D and 3D image classification tasks, comparing its performance against established baselines including DeiT III, RoPE-Mixed, and Vision-Llama. Our findings reveal significant advancements across multiple metrics as compared to the DEIT III basline: LieRE leads to marked relative improvements in accuracy (10.0% for 2D and 15.1% for 3D compared to DeiT). A 3.9-fold reduction in training time for the same accuracy was observed. LieRE required 30% less training data to achieve comparable results. These substantial improvements suggest that LieRE represents a meaningful advancement in positional encoding techniques for multi-dimensional data. The implementation details and reproducibility materials will be made openly available.

## 1 Introduction

While the attention mechanism has achieved widespread use, especially as part of the transformer architecture, attention is invariant to the order of its inputs and requires another mechanism to capture positional information of input tokens Vaswani et al. (2017). This has spurred a line of work in the subarea of positional encodings—methods of encoding positional information in attention mechanisms.

In particular, Rotary Position Encoding (RoPE) has emerged as a technique for encoding relative positional information of text tokens in transformer-based models Su et al. (2024). RoPE's ability to capture relative position information has made it a popular choice for open-source language foundation models such as LLaMA. In particular, RoPE implicitly captures *relative* positions. When the token in position $p_1$ attends to a token in position $p_2$, the effect of RoPE depends on $p_1 - p_2$.

Despite the success of RoPE in sequence tasks Touvron et al. (2023); Chowdhery et al. (2023), it is designed for one-dimensional sequence data. This has resulted in slow adoption for modalities with higher dimensional data, such or data that includes a temporal dimension like videos.

Our work aims to answer whether a single position encoding scheme can work well across both 2D and 3D modalities. If possible, this would enable the use of a simpler common model backbone across a variety of tasks.

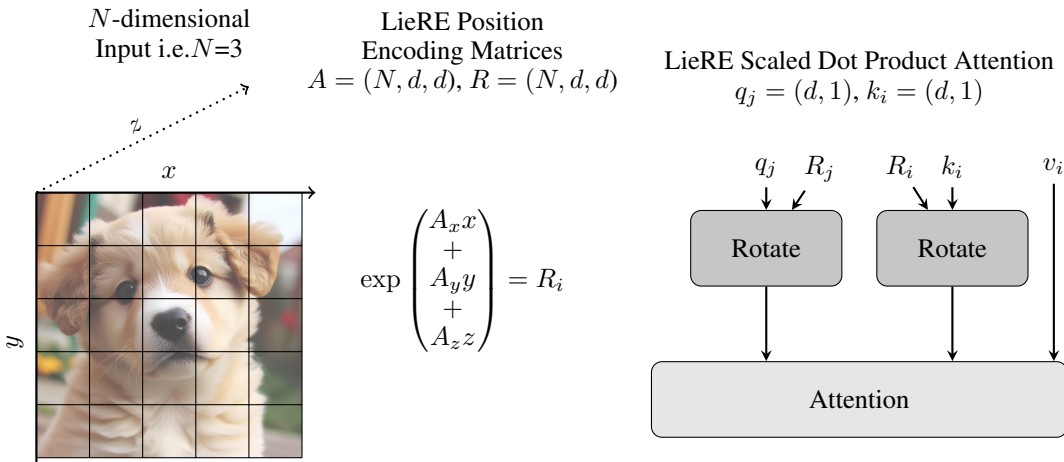

Figure 1: LieRE sketch, where $A$ is a learnable skew symmetric matrix and $R_i = \exp(A[x\ y\ z]) \in \mathbb{R}^{d \times d}$ is the rotation matrix for the first patch in the top left corner of the input with the position $P_i = (x, y, z) \in \mathbb{R}^N$. $d$ is the head dimension. The upper triangle contains learnable LieRE$_\theta$ parameter, while the lower triangle contains their negatives, -LieRE$_\theta$, reflecting the skew-symmetric nature of the matrix.

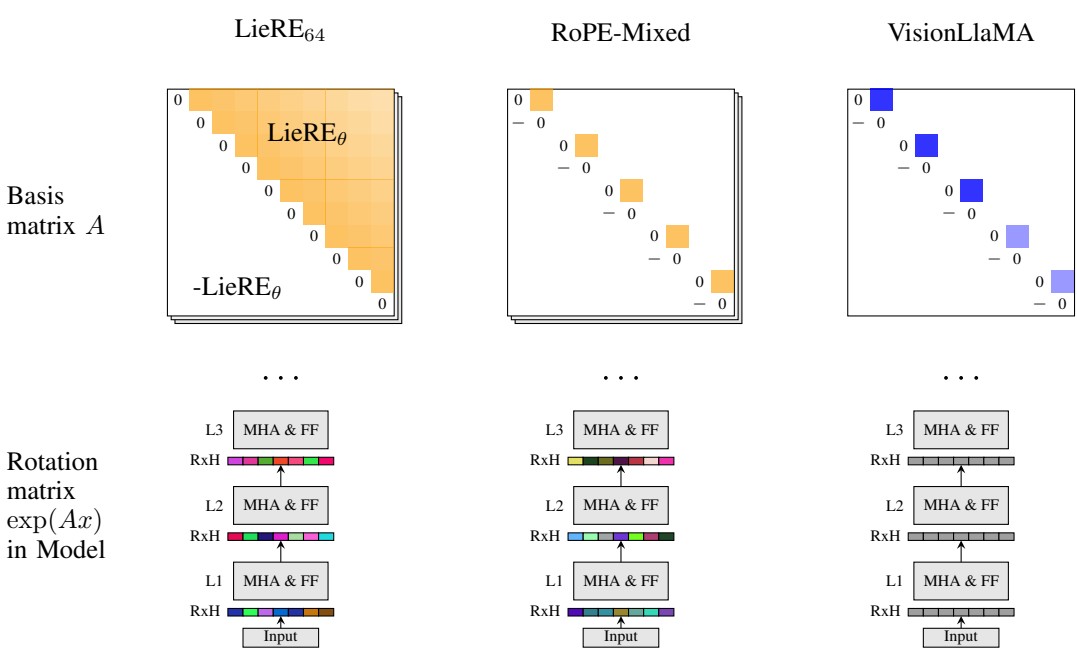

## 1.1 CONTRIBUTIONS

We introduce Lie Relative Encodings (LieRE), a mechanism that allows the attention mechanism to learn how to utilize relative spatial information of its inputs. We show that LieRE is effective on both 2D and 3D inputs of various modalities. Beyond improving classification accuracy, LieRE also reduces the amount of compute and data required during training to achieve a fixed accuracy. On the CIFAR100 task, this translates to 3.9x fewer training steps to achieve the same accuracy as the DeiT baseline and outperforming the DeiT baseline trained on the full set with only a 70% subset of the data. Furthermore, LieRE is simple to implement and adaptable to modalities, requiring only a

tokenizer that also outputs a position in $\mathbb{R}^d$ in addition to a standard embedding. In order to aid the reproducibility of our results we will post our code on github.

## 1.2 RELATED WORK

### 1.2.1 POSITION ENCODINGS

The fact that the original attention mechanism is invariant to the order of tokens has motivated the ongoing development of methods to incorporate positional information into the transformer architecture. We split our literature review into three broad classes of positional encodings: 1) absolute, 2) relative, and 3) contextual.

Absolute encodings generally operate on a per token-level, modifying the embedding of a token to encode the location of the token in the intput. Methods such as sinusoidal and learned absolute encodings add vectors to the input token embedding Vaswani et al. (2017); Devlin et al. (2019); Dosovitskiy et al. (2020). Absolute position measures position with respect to an absolute reference, such as the start of the text or the top left corner of an image.

Relative position encodings instead encode the relative positions of two tokens. One strategy is to learn an embedding for position deltas which can be incorporated into the attention mechanism Shaw et al. (2018); Liu et al. (2021; 2022). However, this incurs quadratic computational cost in terms of the number of tokens. Rotary Position Encodings (RoPE) avoid this cost by rotating the key and query vectors before the attention inner product. The algebraic properties of the block-diagonal rotation matrices used in RoPE ensures only relative positional information is captured in the attention mechanism Su et al. (2024). RoPE is quite widely used in open source LLMs including the PaLM, Llama and Mixtral models Touvron et al. (2023); Chowdhery et al. (2023); Jiang et al. (2024). However, RoPE can perform poorly on inference for larger context sizes than the model was trained on. This has spurred an active line of work extending RoPE to longer contexts, work which we review later.

We refer to the last category of positional encodings as *contextual* position embeddings. This category is defined by encodings that aim to capture semantic positional information lost in traditional absolute and relative position encodings, often motivated by reasoning or mathematical tasks. Contextual Position encodings achieve (CoPE) this by allowing the model to learn how the position is computed Golovneva et al. (2024). Abacus embeddings enable transformers to learn how to handle arithmetic by better exposing the digit structure of numbers McLeish et al. (2024).

### 1.2.2 EXTENSIONS OF ROPE

The efficiency and popularity of RoPE have led to several lines of work building off of it.

One notable one is context extension, which aims to address the fact that RoPE NLP models trained on short documents tend to perform poorly on long documents. Methods like NTK-aware context extension, YaRN and LongRoPE focus on enabling already trained models to handle long context, both with and without finetuning Ding et al. (2024); Peng et al. (2023); Tworkowski et al. (2024); Chen et al. (2023).

Another line of work has been specifically focused on adapting RoPE to image tasks. Both VisionLlama and RoPE-Mixed present relative position encodings inspired by RoPE that are able to encode 2D positional data Chu et al. (2024); Heo et al. (2024). The primary difference is that RoPE-Mixed has a learnable component, whereas VisionLlama does not.

## 2 BACKGROUND

### 2.1 LIE GROUPS IN THE CONTEXT OF ATTENTION

In this section, we aim to provide a minimal introduction to Lie groups so that the reader is able to understand the mathematical motivations behind LieRE. Lie groups are well studied, especially in the context of representation theory, and standard texts including Fulton & Harris (2013) are able to provide a more extensive introduction to the subject.

In this context, Lie groups are smooth sets of matrices that are closed under matrix multiplication and inversion. For every Lie group, the matrix exponential provides a smooth bijective map from a subset of $\mathbb{R}^{n \times n}$, also known as the Lie Algebra, to the Lie group. The exponential map is a diffeomorphism and has the following key property for $U, V \in \mathbb{R}^{n \times n}$ close together:

$$\exp(U - V) = \exp(-V + U) \approx \exp(V)^{-1} \exp(U) \tag{1}$$

Both RoPE (in the context of text) and RoPE-Mixed use block-diagonal rotation matrices with 2D rotations as blocks. These form a special Lie group that is commutative, allowing us to strengthen the statement in equation 1 to

$$\exp(U - V) = \exp(U)\exp(V)^{-1} = \exp(V)^{-1}\exp(U). \tag{2}$$

Our work examines the tradeoff between using the stronger property in equation 2 or increased capacity and the weaker property equation 1.

## 2.2 ATTENTION MECHANISM

LieRE is a modification of the standard attention mechanism to introduce positional information, which we review below. The modification we propose is independent of whether we use multiple heads, so we focus on single-headed attention for simplicity.

Let $X \in \mathbb{R}^{n \times d}$ be the set of input embeddings and $W_Q, W_K, W_v$ be learnable matrices. Let $Q = XW_Q, K = XW_K, V = XW_V$ be the keys, queries and values respectively. The outputs are computed as scores $= \frac{QK^\top}{\sqrt{d_k}}$, $\mathcal{W} = \text{softmax}(\text{scores})$ and final outputs $z = \mathcal{W}V$. We let $Q_i$ and $K_i$ denote the $i$th rows of $Q$ and $K$ respectively.

## 3 METHOD

LieRE is a simple modification to the attention mechanism that is presented in algorithm 1. Recall that we assume that positions are $n$-dimensional vectors, a matrix $A$ is skew-symmetric if $A^T = -A$, and that the matrix exponential of a skew-symmetric matrix, call it $A$, is always a high dimensional rotation matrix.

---

**Algorithm 1** LieRE Attention

1: **procedure** LIERE_ROTATIONS$(p, A)$
2:    $d \leftarrow dimension(p)$
3:    **return** matrix_exp $\left( \sum_{i=0}^{p} A_i p_i \right)$
4: **end procedure**

5: **procedure** LIEREATTENTION$(Q, K, V, A)$
6:    $p \leftarrow$ tokenPositions
7:    $R \leftarrow$ LIERE_ROTATIONS$(p, A)$
8:    *// Multiply each key and query vector by the rotation for that token.*
9:    $K_{\text{rot}} \leftarrow$ BATCHMATMUL$(R, K)$
10:    $Q_{\text{rot}} \leftarrow$ BATCHMATMUL$(R, Q)$
11:    Attention $\leftarrow$ softmax $\left( \frac{Q_{\text{rot}} K_{\text{rot}}^T}{\sqrt{\dim(K)}} \right) V$
12:    **return** Attention
13: **end procedure**

---

**Algorithm 2** RoPE Attention

1: **procedure** ROPE$(X, p, d)$
2:    $\theta \leftarrow \frac{1}{10000^{2i/d}}$   $\forall i \in [0, d)$
3:    **for** $i \leftarrow 0$ **to** $d$ **step** 2 **do**
4:      $X_{\text{rot}}^i \leftarrow X^i \cos p\theta^i - X^{i+1} \sin p\theta^i$
5:      $X_{\text{rot}}^{i+1} \leftarrow X^i \sin p\theta^i + X^{i+1} \cos p\theta^i$
6:    **end for**
7:    **return** $X_{\text{rotated}}$
8: **end procedure**

9: **procedure** ROPEATTENTION$(Q, K, V)$
10:    $p \leftarrow$ tokenPositions
11:    $d \leftarrow$ embeddingDimension
12:    $K_{\text{rot}} \leftarrow$ ROPE$(K, p, d)$
13:    $Q_{\text{rot}} \leftarrow$ ROPE$(Q, p, d)$
14:    Attention $\leftarrow$ softmax $\left( \frac{Q_{\text{rot}} K_{\text{rot}}^T}{\sqrt{d}} \right) V$
15:    **return** Attention
16: **end procedure**

---

When encoding positions $p \in \mathbb{R}^n$, LieRE learns a skew-symmetric basis of matrices $\{A_i\}$ for $i \in [n]$. It encodes a position by writing it in this basis, $\sum_{i=0}^{n} p_i A_i$. We then map the resulting skew-symmetric

matrix to a high-dimensional rotation via the matrix exponential. $R(p) = \exp\left(\sum_{i=0}^{n} p_i A_i\right)$. Learning in the space of skew-symmetric matrices allows us to sidestep some of the difficulty that would come from learning on the manifold of rotation matrices.

LieRE uses the rotation matrix computed above to modify the keys and queries of the standard attention mechanism. LieRE's final step is to modify token $i$'s query and keys as $Q_i' = R(p_i)Q_i$ and $K_i' = R(p_i)K_i$. This modifies the score between tokens $i, j$ to be $X_i^T W_Q^T R(p_i)^T R(p_j) W_K X_j$. Recalling that $R^T = R^{-1}$ for any orthogonal matrix $R$ helps illustrate the encoding of relative positions in equation 1. Note that the *only* difference between LieRE and RoPE-Mixed is that the latter constrains the rotations to be block-diagonal with block size two.

We include the psuedocode for LieRE attention in algorithm 1 beside standard RoPE attention (algorithm 2). As a practical matter, we compute the rotation matrices at the start of the forward pass. By default, the skew bases are learned separately for every layer and attention head except in the experimental section focused on sharing parameters across heads and layers.

Adjusting the skew-symmetric basis matrices' block width allows us to incrementally adjust the capacity allocated towards position encoding. We specify the basis block width as a subscript, eg. $LieRE_8$. When not specified, the block size is equal to the head dimension. If we set the block size to 2, we recover RoPE-Mixed Heo et al. (2024).

## 4 EXPERIMENTS

In order to isolate the effect of changing the position encoding, we use a standard[1] transformer backbone modified to be able to switch between relative position encoding types. We use the standard backbone sizes of ViT-Tiny, ViT-B and ViT-L Dosovitskiy et al. (2020). All experiments use RandAugment Cubuk et al. (2020). We avoid using pre-trained weights in order to maximize the comparability of results between methods. In order to ensure a fair comparison, we explicitly avoid tuning hyperparameters with LieRE and use the same default hyperparameters for all experiments (Appendix B). We evaluate two versions of LieRE, distinguished by the basis matrix tile sizes of 64 and 8, referred to as $LieRE_{64}$ and $LieRE_8$, respectively. Notably, a tile size of 2 corresponds to RoPE-Mixed.

### 4.1 DATASETS AND TASKS

Our experiments are designed to evaluate the efficacy of $LieRE_{64}$ and $LieRE_8$ as a position encoding across both 2D and 3D data. We evaluate LieRE on the classification of 2D (images) and 3D (videos) data. For 3D data and ImageNet-1k (2D), we focus on accuracy. For CIFAR-100 (2D), where training is less resource intensive, we also evaluate LieRE's data and training compute efficiency.

#### 4.1.1 2D CLASSIFICATION

For 2D data we evaluate performance on the CIFAR-100 and ImageNet-1k image classification task Krizhevsky et al. (2009); Deng et al. (2009). We partition our evaluation of performance into four parts. In the first part, we examine accuracy across a range of model architectures on both CIFAR-100 and ImageNet-1k and compare to absolute Dosovitskiy et al. (2020) position encoding and recent related work, RoPE-Mixed Heo et al. (2024), VisionLlama embeddings Chu et al. (2024).

In the second part, we take advantage of the relatively modest amount of compute resources necessary to train a model for CIFAR-100 to examine LieREs impact on data efficiency. We also measure training *compute* efficiency by comparing the number of training steps necessary to achieve a fixed level of validation accuracy.

In the third part, we evaluate the impact of LieRE with various scales. We vary the capacity of the transformer backbone with corresponding to ViT-T, ViT-B and ViT-B, as proposed in Dosovitskiy et al. (2020), the number of learned LieRE skew-symmetric basis (one, per-attention-head and per-layer) and the LieRE basis capacity by imposing a block diagonal structure on the Lie algebra that allows us to vary the added capacity.

---

[1]https://github.com/kentaroy47/vision-transformers-cifar10

Table 1: 2D image and 3D video classification Top-1 Accuracy (95% confidence intervals) results. All models use 85.1M parameters for 2D tasks and 88.7M parameters for 3D task Krizhevsky et al. (2009); Deng et al. (2009); Soomro et al. (2012); Stein et al. (2019) * equivalent to DeiT, ** equivalent to Vivit (spatio-temporal).

| Method | CIFAR-100 | ImageNet-1k | UCF101 |
|---|---|---|---|
| Abs. Pos. E.*,** | 63.9 (62.9-65.8) | 66.1 (65.7-66.5) | 40.9 (40.5-41.3) |
| VisionLlama RoPE | 65.5 (64.6-66.5) | 65.4 (65.0-65.8) | 45.0 (44.6-45.4) |
| RoPE-Mixed | 68.8 (67.9-69.7) | 68.8 (68.4-69.2) | 46.3 (45.9-46.7) |
| $LieRE_8$ | **70.3 (69.4-71.2)** | XXX (XXX-XXX) | **47.0 (46.6-47.4)** |
| $LieRE_{64}$ | 70.0 (69.1-70.9) | **69.3 (68.9-69.7)** | 44.7 (44.3-45.1) |

In the fourth part, we measure how much different models depend on the positional information in the image/video by shuffling the patches. A higher accuracy drop with randomly shuffled patches means the model relies more on the positions of the patches during inference.

### 4.1.2 3D CLASSIFICATION

In this section, we introduce Rotary Position Encodings for 3D data and compare LieRE-based transformers with transformers with RoPE-Mixed and absolute encoding similar to the previous section Arnab et al. (2021); Heo et al. (2024). For the 3D experiments, we examine video classification performance in the UCF101 dataset Soomro et al. (2012). Again, we did not optimize any hyperparameters for the LieRE model and used the dataloader from Tong et al. (2022). The full set of hyperparameters may be found in appendix B.

### 4.1.3 MULTI-RESOLUTION CLASSIFICATION

In this section we compare the ability of methods to generalize to image resolutions not seen during training. We evaluate two training recipes inspired by Heo et al. (2024). The first recipe matches the rest of the paper and consists of training the models on images of size $224 \times 224$ for 200 epochs. The second adds an additional fine-tuning step at size $256 \times 256$ for 30 epochs. The full details can be found in appendix B.

## 5 RESULTS

### 5.1 ACCURACY

For 2D image classification tasks, we demonstrate that the LieRE-based transformer achieves a relative performance improvement in accuracy of 10.0% over DeiT Touvron et al. (2022), 7.3% over RoPE adaptation in VisionLlama Chu et al. (2024), and 2.2% over RoPE-Mixed Heo et al. (2024) on CIFAR-100, with similar accuracy trends observed on ImageNet Deng et al. (2009) (table 1).

For 3D input classification tasks using the UCF101 dataset Soomro et al. (2012), we observe a relative accuracy improvement of the LieRE-based transformer of up to 15.1% compared to absolute position embeddings and at least 1.5% compared to RoPE-inspired position encodings (table 1).

We further evaluate the accuracy of our model on the ImageNet validation set across varying inference resolutions. Specifically, we scale the input images to resolutions of $196 \times 196$, $256 \times 256$, $320 \times 320$, $384 \times 384$, and $448 \times 448$ pixels per dimension, and present the resulting accuracies in figure 2.

For position assignment, we adopt a sequential approach where token positions are scaled proportionally to the image dimensions. For example, doubling the length of an image in each dimension doubles the range of positional indices. This method outperforms rescaling positions to a fixed range, as demonstrated by superior results for both RoPE-Mixed and LieRE across the evaluated training recipes.

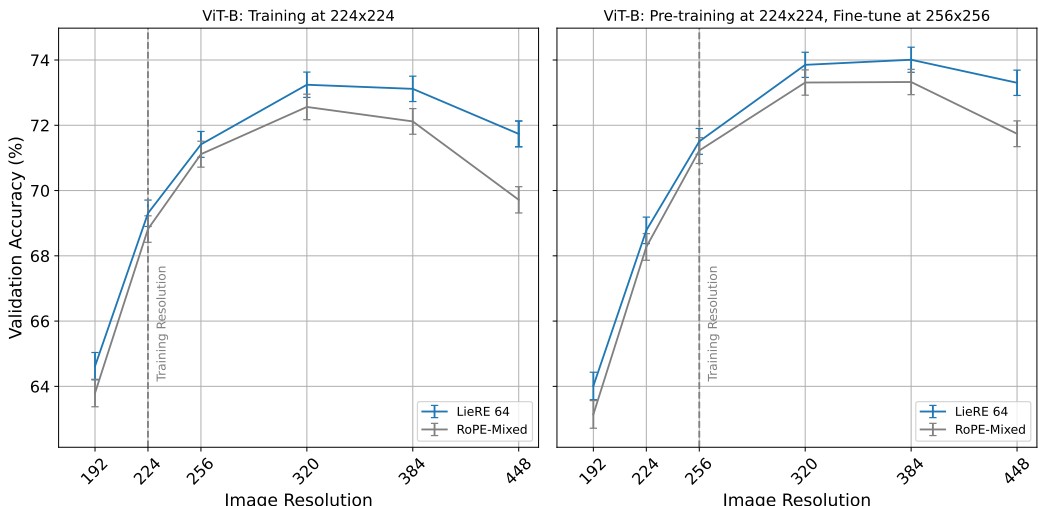

Figure 2: LieRE and RoPE-Mixed Accuracies at various resolutions on ImageNet.

## 5.2 Data efficiency

We further observe that learnable relative position encodings, such as LieRE and RoPE-Mixed, exhibit substantially greater data efficiency compared to prior transformer methods for 2D image classification on the CIFAR-100 dataset (figure 3b).

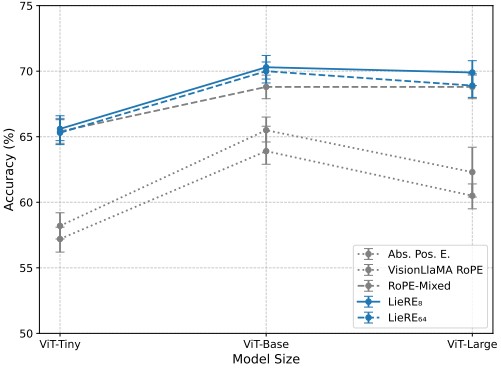

Figure 3a: Performance behavior on CIFAR-100 (2D Image Classification) over ViT-Tiny (22M), ViT-Base (85M), ViT-Large (302M) for LieRE Rope-Mixed and Absolute Encoding (Appendix, table 4).

Figure 3b: Data ablation on for different position embeddings on CIFAR-100.

## 5.3 Model Scaling

We investigate three dimensions of capacity scaling: transformer backbone parameters, LieRE basis parameters, and the use of distinct LieRE bases across heads and layers.

### 5.3.1 Transformer backbone capacity

Additionally, we analyze the impact of incorporating LieRE on performance across different model sizes on the CIFAR-100 dataset, as shown in table 4. Our results demonstrate that $LieRE_8$ consistently outperforms alternatives across all evaluated model sizes.

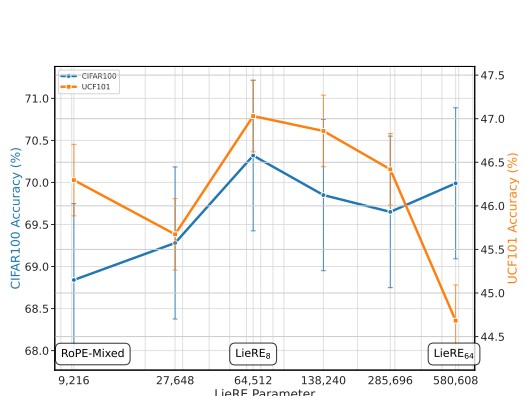

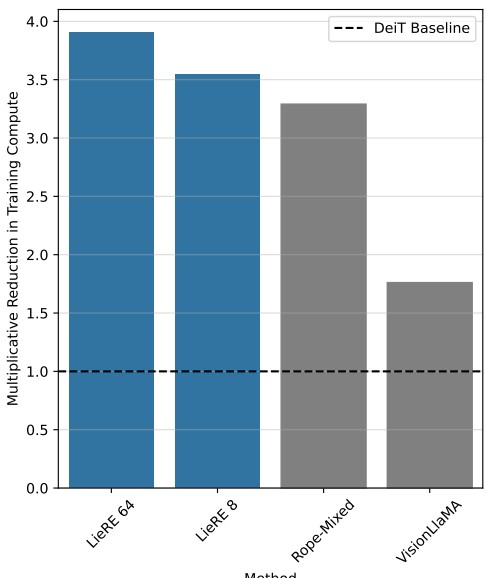

Figure 4a: Performance varies with skew-symmetric basis learnable dimensions, referred to as LieRE parameters. This is equivalent to increasing the tile size in the skew-symmetric basis ($2 \times 2, 4 \times 4, 8 \times 8, 16 \times 16, 32 \times 32, 48 \times 48$). For both 2D (CIFAR-100) and 3D (UCF101) LieRE with tile size $8 \times 8$ performs superior.

Figure 4b: The LieRE spatial encoding allows the model to match the performance of absolute position encodings with substantially less training time. Please fine the learning curves in the appendix 5.

Table 2: Accuracy with parameter sharing over heads and layers for ViT-B sized models on CIFAR-100.

| FLOP | All Shared | Shared Across Layers | Shared Across Heads | RoPE-Mixed | LieRE$_{64}$ | LieRE$_8$ |
|---|---|---|---|---|---|---|
| 5.684G | | ✓ | ✓ | 68.8 | 70.0 | 70.3 |
| 5.684G | | ✓ | | 68.7 | 69.5 | 69.8 |
| 5.613G | | | ✓ | 69.5 | 69.7 | 69.7 |
| 5.613G | ✓ | | | 68.3 | 69.4 | 69.5 |

### 5.3.2 BASIS PARAMETERS

LieRE adds a small amount of capacity to the model ($580k$ parameters for the ViT-B backbone we use for most experiments), leading to the natural question of how helpful the marginal capacity is.

We vary capacity by enforcing a block diagonal structure on the skew-symmetric basis. Varying the block size allows us to approach LieRE$_{64}$. Recall that using $2 \times 2$ blocks recovers exact commutativity and is referred to as RoPE-Mixed. In figure 4a we evaluate accuracy versus block dimension.

### 5.3.3 IMPACT OF SHARING LIERE PARAMETERS ACROSS HEADS AND LAYERS

Table 2 presents the impact of sharing LieRE parameters across attention heads and layers on CIFAR-100 classification performance for RoPE-Mixed, LieRE$_{64}$ and LieRE$_8$ . We evaluate whether learning separate positional encodings for each attention head and layer provides performance benefits. We observe that learning across heads and layers yields superior performance followed by learning across layers.

### 5.4 COMPUTE EFFICIENCY

Training transformers can necessitate substantial computational resources, which can hinder equitable access to research and development of machine learning methods. We demonstrate that the LieRE-based transformer requires 3.9 times less training epochs on CIFAR-100 to achieve comparable

Table 3: Relative accuracy drop for 2D image classification (CIFAR-100) and Video recognition (UCF101) after patch shuffling

| Method | CIFAR-100 (2D) | | | UCF101 (3D) | | |
|---|---|---|---|---|---|---|
| | Before Shuffling↑ | After Shuffling↓ | Drop(%) ↑ | Before Shuffling↑ | After Shuffling↓ | Drop(%) ↑ |
| Abs. Pos. E. | 63.9 | 19.6 | 69.3 | 40.9 | 39.5 | 0.0 |
| VisionLlama RoPE | 65.5 | 29.7 | 54.8 | 45.0 | 37.0 | 17.7 |
| RoPE-Mixed | 68.8 | 17.1 | 75.1 | 46.3 | 28.2 | 39.1 |
| LieRE$_8$ | 70.3 | 12.3 | 82.5 | 47.0 | 27.8 | **40.9** |
| LieRE$_{64}$ | 70.0 | 10.8 | **84.6** | 44.7 | 28.0 | 37.4 |

performance to the Absolute Position Embedding baseline (as used in DeiT III Touvron et al. (2022)). This represents the largest reduction in training time compared to recent works such as VisionLlama and RoPE-Mixed Chu et al. (2024); Heo et al. (2024). We note that this is the first training efficiency comparison of these recent methods. figure 4b shows the amount of allowable compute reduction to achieve the same accuracy achieved by absolute position encodings (DeiT baseline) after 200 epochs. LieRE demonstrates the largest win, allowing a 3.9X reduction in training compute to achieve the same accuracy.

### 5.5 PATCH SHUFFLING

Shuffling patches and frames allows us to see how much the model is able to use the positional information in its inputs. A model whose architecture does not allow/encourage the use of positional information will converge to a representation similar in spirit to a bag-of-words, where the relative locations of pixels/voxels do not matter. A greater dropoff in accuracy during shuffling is indicative that the model more heavily utilizes positional information.

We evaluate models using the decline in accuracy when evaluating on shuffled patches. We observe the most significant decline LieRE-based transformers, leading to the conclusion that LieRE models are most capable at using positional information. The complete results are displayed for CIFAR-100 and table for UCF101 (table 3) .

## 6 LIMITATIONS

While LieRE shows promising results across multiple modalities and input dimensionalities, there are a few limitations worth noting. Our method is specifically designed to modify the inner product, making it compatible with most attention schemes, including original attention and linear attention. However, this specificity may limit its applicability to other architectures, like convolutional neural networks, that do not rely on inner product-based attention mechanisms. Future work could explore adaptations of our method to a wider range of architectures.Furthermore, the current formulation of our method is designed to encode vector positions in $\mathbb{R}^d$. While this is sufficient for many applications, it may not be directly applicable to tasks that require encoding poses in $SE(3)$ like robotics. Further research may be necessary to adapt our method to effectively handle such representational requirements.

Despite these limitations, we believe that our work provides much-needed insight into how to improve model performance and reduce training costs by encoding relative position information across various modalities and input dimensionalities.

## 7 BROADER IMPACTS

LieRE, our proposed method for encoding positional information in attention mechanisms, has demonstrated improvements in 2D image and video classification tasks. We are particularly excited about how LieRE can expand the applicability transformers to $n$ dimensional inputs. This may

lay down the tracks to apply that same relative position encoding within and across modalities and settings.

## 8 CONCLUSION

In this paper, we introduced Lie group Relative position Encodings (LieRE), a position encoding that can effectively encode relative position information for attention mechanisms across modalities and input dimensionalities. Through experiments on 2D image classification (CIFAR-100, ImageNet-1k) and 3D video classification (UCF101), we demonstrated that LieRE achieves better performance compared to existing positional encoding methods. Beyond improving accuracy, LieRE also exhibits data and compute efficiency. On CIFAR-100, LieRE requires 3.5 times less training computed to match the performance of the baseline model with absolute position encodings. Furthermore, LieRE can outperform the baseline trained on the full dataset while using only 70% of the training data, highlighting its data efficiency. The key advantages of LieRE include its simplicity, flexibility, and ease of adaptation to new modalities. By requiring no changes to the tokenizer other than outputting positions and no other code changes, LieRE may provide a unified and efficient approach for transformers to process and learn from various data modalities within a single architecture.

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

# A  Appendix

# B  Experiment Hyperparameters

The backbone for all experiments is configured as ViT-B, with 12 layers, a hidden dimension of 768, and an intermediate dimension of 3096. We use a dropout of 0.1. We used CLS pooling in our implementation to facilitate comparability with existing literature in the field. Further experiments revealed substantial performance improvement with mean pooling and LieRE. We use the pytorch lightning framework for all experiments Falcon (2019).

## B.1  2D Image Classification

The CIFAR experiments where trained on 8xL4 GPUs with 24GB of VRAM each and all took under 30 minutes to complete. The basis capacity scaling experiment was conducted using RTX6000 GPUs. The ImageNet experiments were trained on 8xL40 GPUs and all took less than 2 days and 5 hours of runtime including time lost due to preemption and resource sharing. We use a cosine learning rate schedule with an initial learning rate of $1E-4$ and train for 200 epochs. We use an effective batch size of 512. We use a patch size of $4 \times 4$ on the original $32 \times 32$ image for CIFAR-100 and a patch size of $16 \times 16$ on the randomly cropped and resized $224 \times 224$ image. All vision experiments used RandAugment Cubuk et al. (2020). We use the ADAM optimizer with betas of 0.9 and 0.999 and $\epsilon = 1e - 8$. The hyperparameters were tuned with RoPE-Mixed and selected before conducting the LieRE trainers as to ensure a fair comparison.

## B.2  3D Video Classifications

The 3D classification experiments were conducted on either $8 \times A100$ 40GB GPUs or $4 \times A100$ 80GB GPUs with the effective batch size held constant either by using a gradient accumulation or increasing the batch size. Similar to 2D classification, we use an initial learning rate of $1E-4$ with a cosine decay, trained for 200 epochs, and had a total batch size of 64 and a patch size of $2 \times 16 \times 16$ on the randomly cropped and resized $8 \times 224 \times 224$ video/image. We use the ADAM optimizer with betas of 0.9 and 0.999 and $\epsilon = 1e - 8$.

## B.3  Multi-resolution Classification

The second training recipe consists of 30 epochs with an initial learning rate of 1E-5 with a cosine decay. This mirrors the DEIT III training reciple that first pretrains at a lower resolution and finetunes at a higher resolution.

## B.4  CIFAR-100 Performance Across Model Scales

We also evaluate how the inclusion of LieRE affects performance across model sizes on the CIFAR-100 in table 4. We observe that LieRE retains a statistically significant lead in performance across all three model sizes.

### B.4.1  Basis parameters

Table 4: Comparison of Position Encoding Methods for Different ViT Models Sizes on CIFAR-100, Accuracy (bootstrapped 95%CI)

| Position Encoding | ViT-Tiny | ViT-Base | ViT-Large |
|---|---|---|---|
| Abs. Pos. E. | 57.2 (56.2-58.1) | 63.9 (62.9-65.8) | 60.5 (59.5-61.4) |
| VisionLlaMA RoPE | 58.2 (57.2-59.2) | 65.5 (64.6 -66.5) | 62.3 (60.4-64.2) |
| RoPE-Mixed | 65.4 (64.5-66.4) | 68.8 (67.9-69.7) | 68.8 (67.9-69.7) |
| LieRE$_8$ | 65.6 (64.7-66.6) | 70.3 (69.4-71.2) | 69.9 (68.9-70.8) |
| LieRE$_{64}$ | 65.3 (64.4-66.3) | 70.0 (69.1-69.7) | 68.9 (68.0-69.8) |

## B.5 PYTHON IMPLEMENTATION OF LIERE ROTATION MATRIX COMPUTATION

```python
basis_raw_params = nn.Parameter(
    torch.rand(
        input_dimensionality,
        head_dim,
        head_dim,
    ) * 2 * math.pi # optional, inspired from RoPE-Mixed paper
)

upper_triangle = (
    torch.triu(basis_raw_params, diagonal=1)
)
skew_bases = upper_triangle - torch.transpose(upper_triangle, -1, -2)
in_basis_positions = (
    positions.reshape(list(positions.shape) + [1] * 2) * skew_bases
)
rotation_log = torch.sum(in_basis_positions, dim=-3)
rotation = torch.matrix_exp(rotation_log.to(dtype=torch.float32)).to(
    dtype=positions.dtype)
```

## B.6 FLOPS COMPARISON OF METHODS

We find that since all methods we examine introduce a computational cost that is at most linear in the number of tokens, and runtime is dominated by the quadratic attention component, there is no substantial difference in computational efficiency between the methods. We list inference FLOP of the various methods in table 5.

Table 5: FLOP analysis with percentage increase compared to absolute position encodings

| Position Enc. | ViT-Tiny (22M) | ViT-Base (85M) | ViT-Large (302M) |
|---|---|---|---|
| Abs. Pos. E.* | 0.963G | 5.607G | 19.856G |
| VisionLlaMA RoPE | 0.963G (+0.001%) | 5.607G (+0.002%) | 19.856G (+0.000%) |
| RoPE-Mixed | 0.964G (+0.104%) | 5.609G (+0.036%) | 19.863G (+0.035%) |
| LieRE$_8$ | 0.968G (+0.519%) | 5.617G (+0.178%) | 19.882G (+0.065%) |
| LieRE$_{64}$ | 0.970G (+0.727%) | 5.684G (+1.375%) | 20.061G (+1.033%) |

## B.7 VALIDATION LOSSES

Table 6: 2D image and 3D video classification Top-1 Validation loss (95% confidence intervals) results. All models use 85.1M parameters for 2D tasks and 88.7M parameters for 3D task Krizhevsky et al. (2009); Deng et al. (2009); Soomro et al. (2012); Stein et al. (2019) * equivalent to DeiT, ** equivalent to Vivit (spatio-temporal).

| Method | CIFAR-100 | ImageNet-1k | UCF101 |
|---|---|---|---|
| Abs. Pos. E.*,** | 1.56 (1.47-1.56) | 1.84 (1.81-1.86) | 2.94 (2.92-2.96) |
| VisionLlama RoPE | 1.56 (1.51-1.61) | 1.98 (1.94-2.01) | 2.66 (2.63-2.69) |
| RoPE-Mixed | 1.38 (1.33-1.43) | 1.72 (1.68-1.74) | 2.52 (2.49-2.54) |
| LieRE$_8$ | **1.36 (1.31-1.41)** | **XXX (XXX-XXX)** | **2.47 (2.44-2.49)** |
| LieRE$_{64}$ | 1.37 (1.33-1.42) | 1.73 (1.70-1.76) | 2.64 (2.62-2.67) |

### B.8 BASIS PARAMETERS

Table 7: Accuracy Results for Different LieRE$_\Theta$ Parameters

| Dataset | LieRE$_\Theta$ Parameter | Tile Size | Accuracy (%) | CI (95%) |
|---------|--------------------------|-----------|--------------|----------|
| CIFAR100 | 9216 | 2 | 68.84 | (67.93–69.75) |
| CIFAR100 | 27648 | 4 | 69.28 | (68.38–70.18) |
| CIFAR100 | 64512 | 8 | 70.32 | (69.42–71.22) |
| CIFAR100 | 138240 | 16 | 69.85 | (68.95–70.75) |
| CIFAR100 | 285696 | 32 | 69.65 | (68.75–70.55) |
| CIFAR100 | 580608 | 64 | 69.99 | (69.09–70.89) |
| UCF101 | 9216 | 2 | 46.30 | (45.89–46.71) |
| UCF101 | 27648 | 4 | 45.67 | (45.26–46.08) |
| UCF101 | 64512 | 8 | 47.03 | (46.62–47.44) |
| UCF101 | 138240 | 16 | 46.86 | (46.45–47.27) |
| UCF101 | 285696 | 32 | 46.42 | (46.01–46.83) |
| UCF101 | 580608 | 64 | 44.68 | (44.27–45.09) |

### B.9 VALIDATION ACCURACY CURVES

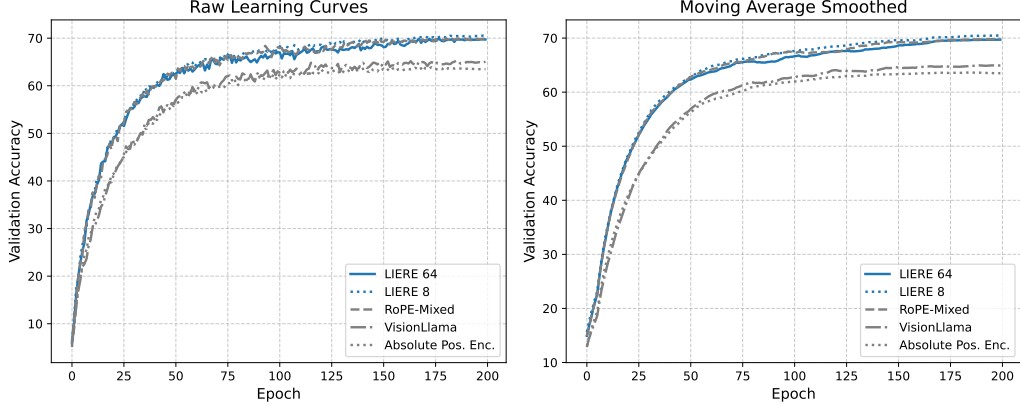

Figure 5: CIFAR-100

## B.10 TRAINING LOSS CURVES

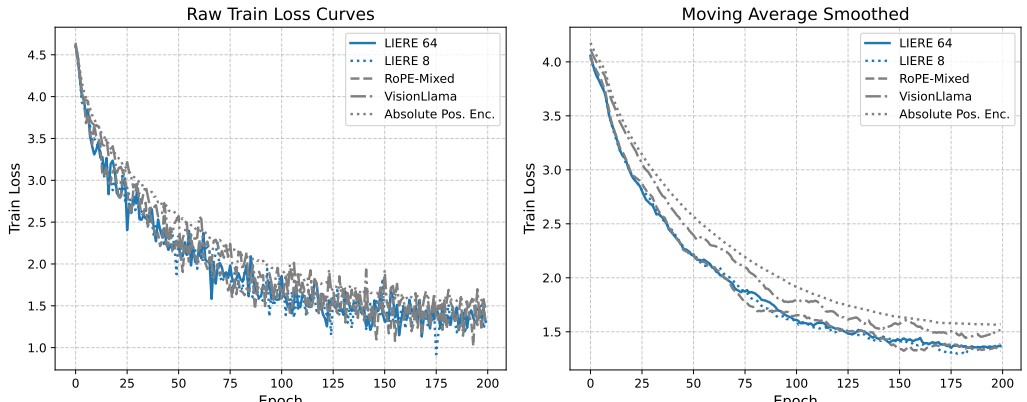

Figure 6: CIFAR-100

