# OpenReview forum: "LieRE: Generalizing Rotary Position Encodings to Higher Dimensional Inputs"
_ICLR.cc/2025/Conference — Submitted to ICLR 2025_

### Official Review · Reviewer_BRzg · 2024-11-01

**Soundness:** 2
**Presentation:** 2
**Contribution:** 2
**Rating:** 3
**Confidence:** 4

**Summary:**

The paper proposed an extension to the prior work RoPE attention. The central idea is to extend the 1-d position encoding in RoPE to handle higher-dimension position indices. In RoPE, the feature vectors are rotated before the attention inner products to take into account the distance between two tokens. For each feature vector of d-dimension, d/2 rotations in 2d space were used. The authors argue that using 1 rotation of d-dimension space, taking into account multiple token indices, is better.

Lie theory is used to map position indices to "rotations", via the exponential map.

**Strengths:**

The use of Lie groups for position-aware attention is a novel and interesting idea.

**Weaknesses:**

- The method is not clearly defined, with proper mathematical description. By code segment in the appendix, is also not explained, particularly not in a way that connects with the main text.
- It is not intuitive why a single rotation is better than multiple rotations. Actually, multiple rotations of RoPE have the benefit of adaptively capturing different levels of distance effects.

**Questions:**

- Please define the A matrix in detail. Clarify the notation n and d in Section 3.2 (use them consistently please).
- Can you conduct experiments on a data set with varying image sizes and resolutions? In other words, a single dataset with different scales of spatial correlation.

---

> ### Author Response · Authors · 2024-11-27
>
> **Explanation of method:** We updated the method section based on your feedback. We replicate a compressed version below for your convenience.
>
> A matrix is skew symmetric if $A^T=-A$, and any matrix $X$ can be made skew symmetric using `torch.triu(x, diagonal=1) - torch.triu(x, diagonal=1)^T`. In order to encoding positions in a $d$-dimensional space LieRE learns sets of $d$ skew-symmetric matrices which we refer to as the skew-symmetric basis. Recall that in a typical ViT, there is a 1:1 mapping between tokens and patches, leading to each patch having a unique position. A token’s position vector $p$ is then encoded as a rotation matrix $R(p) = \exp{\left(\sum\limits_{i=0}^d A_i p_i\right)}$, where $\exp$ refers to the matrix exponential. Then the individual attention head’s keys and queries are modified with a tensorized batch matrix multiplication that rotates each token’s keys and queries by the rotation matrix for that token.
>
> Outside of the section where we experiment with sharing parameters, this is done with separate skew-symmetric bases for each attention head in every layer.
>
> We are also updating the algorithm box to reflect this greater level of detail.
>
> **Why a single high n-dimensional rotation does better than $n/2$ 2D rotations:** Using a single high dimensional rotation actually increases the capacity of the mode compared to using multiple  2D rotations. Q$n/2$ 2D rotations only have $n/2$ degrees of freedom, whereas an $n$-dimensional rotation has $n(n-1)/2$ degrees of freedom. These additional degrees of freedom allow the model to learn a richer set of positional encodings. See the “Motions and dimensions” section [here](https://en.wikipedia.org/wiki/Degrees_of_freedom_(mechanics)) for a reference. This could also be seen as a corollary of skew-symmetric matrices being the Lie algebra of the high dimensional rotation matrix Lie group. It is important to keep in mind that applying a 2D rotation to coordinates 1,2, another 2D rotation to coordinates 3,4 etc, is a special case of a single high dimensional rotation meaning the LieRE can represent RoPE-style methods exactly in addition to being able to represent high dimensional rotations which cannot be broken down into a set of 2D rotation matrices on the block diagonal.
>
> **Experiments with varying image sizes and resolutions:** This is an excellent suggestion. We evaluated both LieRE and RoPE-Mixed on different resolutions of the imagenet validation set below.
>
> Please see the results at https://imgur.com/a/bnPmERw
>
> We evaluate the methods under two training recipes. The first is the original 200 epoch imagenet training recipe used throughout this paper that always trains at a resolution of 224x224. The second one mirrors the DEIT-III recipe and adds an additional 30 epochs at a 240x240. All models perform better with the two-stage training recipe. The performance benefit of LieRE increases as the resolution increases.
>
> This performance increase is particularly exciting in the context of RoPE-Mixed having already shown large improvements over absolute position encodings.
>
> Please let us know if there is anything else we can do to improve your rating. Your feedback has already been very helpful and led to the exciting multi-resolution results.

---

### Official Review · Reviewer_EcnH · 2024-11-02

**Soundness:** 3
**Presentation:** 4
**Contribution:** 4
**Rating:** 10
**Confidence:** 5

**Summary:**

This paper proposes a symmetry-aware method for positional encoding (LieRE) that outperforms the state-of-the-art methods (RoPE), by considering subspace rotations of dimension higher than $2$ in the self-attention's dot-product (the orthogonal symmetry group action given by $(R,(q,k))\mapsto (Rq,Rk)$). Using a Lie group theory the resulting rotation encoding is deduced as $R=\exp(\sum_iA_ix_i)$ where $i$ is each domain dimension and $A_i$ is learnable skew-symmetric matrices. Furthermore, shuffling patches causes more accuracy drop in LieRE than RoPE, which verifies that the model relies more on positional encodings by using LieRE. It also outperforms existing methods in terms of training time and dependency on dataset size.

**Strengths:**

- The positional encoding structure is fundamentally important in Transformer architectures. Improving over the state-of-the-art has a great impact.
- I find it particularly interesting to treat the spatial and embedding dimensions of the hidden states equally, which I believe results in a unified understanding of the embedding space as a tensor field. Lie group generalizes rotary positional embedding from dimension $2$ to $n$ by regarding RoPE as a commutative special case.
- I find it interesting that sharing parameters across layers (LieRE-Commute over RoPE-mixed) improve learnable positional encodings.

**Update**: Thanks for adding the experiments to validate LieRE’s performance. I have raised the score from 8 to 10, the presentation score to 4, and confidence to 5. Good luck.

**Weaknesses:**

1. Although the relative gain over existing methods is fair and remarkable, <70% accuracy on CIFAR100 and ImageNet and ~50% on UCF101 is far from optimal. For example, the referred paper (Heo et al. 2024) reports >80% accuracy. It would be more convincing to improve the baseline.
2. The 3.5x reduction in training time is compared under the wall time of 200 epochs, which means the same performance is obtained at around 57 epochs for LieRE. I wonder how these methods compare in terms of the best test loss, and the converged training loss (which means after 200 epochs). Running longer experiments may also help remedy poor baselines.
3. I find the compute efficiency less informative than the learning curve. The FLOPs analysis is of practical interest but looks trivial since positional encoding is a lightweight part of the model.

Minor issues:
- Table 2: I find the word "stem" in Table 2 confusing and unnecessary. Clarifying it in the text rather than just in Figure 3a would help.
- Many \citet should be \citep
- Table 2 line 381: Rope should be {RoPE}.
- Line 400: Figure ??

**Questions:**

1. I don't understand why commutativeness in the RHS of Equation (2) is not $\exp(V)^{-1}\exp(U)$, but $\exp(U)^{-1}\exp(V)$. Is it a typo?
2. Should line 201 "We present attention with LieRE and LieRE-Commute in Algorithm 1 and 2" reverse the order?
3. Would it be better to clarify the relation between "LieRE-Commute, learnable RoPE and RoPE-mixed" already here in the method section (now this does not appear until 5.3.3), and remind the reader in the background why the group $SO(d)$ not abelian if $d>2$, preferably with some one-line intuitions?
4. Do you have more intuition to justify the reason why the stem option outperforms layer or head heterogeneous implementations? For example, does it suffice to think of the $O(n)$ symmetry in the dot product that only needs one representative element in the quotient group, instead of striving to learn it repeatedly in every layer or head subspace?
Based on this, would you think it is possible to unify the representation by rotationally aligning the subspaces of each layer?

---

> ### Author Response · Authors · 2024-11-27
>
> Thank you for the careful review and constructive feedback of the manuscript.
>
> **Stronger baseline:** The shorter training recipe was motivated by more limited computational resources. We are also cautiously optimistic that we will have results under the DEIT III training recipe used by Heo et al by the deadline, but that depends on how much GPU time we get between now and then. We will share the results once it completes. In the meantime, we fine tuned the models for an additional 30 epochs as part of the multi-resolution classification task, closing the gap somewhat with the much longer training recipe.
> *Update on 12/2: We are on epoch 346/400 of the longer recipe pretraining and are hoping to be able resume the job soon.*
>
> **Detailed FLOPs information:** Thanks for the feedback. We moved the detailed FLOPs table to the appendix in order to make room for the learning curves and multi-resolution classification results. We also added a table to the appendix with the best test losses for all results.
>
> **Clarity:** For table 2 we updated the column names to “All Shared”, “Shared Across Heads”, “Shared Across Layers” and corrected Rope to RoPE. We corrected the commutativity typo in equation 2, corrected the usage of citet and citep and fixed ??? references. We also reversed the order of algorithms 1 and 2 in line 208. We moved the relationship between RoPE-Mixed and Liere into the method section (section 3).
>
> **Performance of sharing LieRE parameters across heads and layers:**
>  We attribute this to the memory issue with further details in the official comment.
>
> **Test losses and learning curve**: We added the validation losses and learning curves to the appendix. The losses and accuracy curves are well converged. We also trained LieRE on imagenet for 400 epochs and could not see substantial gains.
>
> Thank you again for the review! Please let us know if there is anything else we can do to improve the paper as we await the longer training run.

---

### Official Review · Reviewer_CJWD · 2024-11-09

**Soundness:** 3
**Presentation:** 1
**Contribution:** 3
**Rating:** 3
**Confidence:** 3

**Summary:**

The authors propose a position encoding technique that can be used with attention mechanisms for 2D and 3D data. In contrast to RoPE which learns a block diagonal (2x2 blocks) rotation matrix transformation of key and query matrices from relative position information, LieRE learns a general rotation matrix transformation of key and query matrices from absolute position information. While this introduces additional parameters, the authors mitigate this by sharing parameters across attention heads. The authors show that with the combination of these strategies, LieRE improves predictive performance, training efficiency and data efficiency.

**Strengths:**

The authors propose a novel position encoding scheme with improved performance over baseline models. The improved performance is in terms of predictive performance, training efficiency, and data efficiency. Moreover the model can be used for 2D and 3D data.

**Weaknesses:**

The writing could be improved. The document would benefit from additional proofreading. For example, in several places (Lines 169, 173) the text reads ‘equation equation’; ‘figure’ and ‘table’ should be capitalized (Line 187, 397); and the notation for updated keys and queries is inconsistent (Line 188, 189, 209). The clarity of the document would be improved. For example, it would benefit the reader if the equation for the attention mechanism were provided 3.1; some text describing the algorithm would benefit the reader; it would be nice to show that LieRE-Commute and RoPE-mixed are special cases of LieRE.

The organization could be improved. Using subsubsections in the related work doesn’t seem necessary, and takes up space that might be used to clarify the method section. Some details of the method are presented for the first time in the Results section (e.g., that the parameters of LieRE and its variants are shared across attention heads). Figures are often far from where they are referenced.

**Questions:**

* When the authors say ‘generator space’ do they mean Lie algebra?
* In the definition of R_{LieRE} consider using A(x) as the argument to the exponential map
* (Line 400) Broken figure reference
* Do the authors have thoughts on why LieRE improves training/data efficiency
* Should the final statement in eq 2 be exp(V)^-1 exp(U)?

---

> ### Author Response · Authors · 2024-11-27
>
> Thank you for taking the time to review our paper and providing feedback that we were able to use to improve the manuscript.
>
>
> **Style and Proofreading:** Corrected issues such as "equation equation" (Lines 169, 173), inconsistent notation (Lines 188, 189, 209), and improper capitalization of "Figure" and "Table." Used A in argument to exponential to clarify that the input is skew symmetric. Fixed commutativity typo in equation 2. Additional proofreading improved overall clarity and eliminated typos.
>
>
> **Clarity:** Added background section for attention that includes equations. Extensive improvements to the writing in section 3 (method), expanded algorithm descriptions, and clarified how RoPE-Mixed are special cases of LieRE.
>
>
> **Organization:** Reduced usage of subsubsections, moved details on parameter sharing to the method section, and repositioned figures, tables and algorithms closer to their references.
>
>
> **Why LieRE improves efficiency:** LieRE enables the model to depend much more on the token positions. We refer to the patch shuffling experiment as evidence of this. Furthermore, dropping the commutativity allows the model to encode both absolute and relative positional information unlike prior methods that often focused on one or the other. The RoPE-Mixed paper also showed some improvement from using RoPE-Mixed combined with absolute position encodings. LieRE lets one combine both absolute and relative positional information in a consistent way.
>
>
> Thank you again for your constructive feedback that helped improve the paper. Please let us know if there is anything else we can do to improve the paper and score.

---

### Author Response · Authors · 2024-11-27

We sincerely thank the reviewers for their thoughtful feedback and for taking the time to evaluate our submission.

In response to the reviewers’ suggestions, we have made several significant improvements to the manuscript.

**Resolution Generalization Experiments:** As per reviewer 3’s suggestion, we have added experiments to evaluate the method’s generalization across resolutions. LieRE performs strongly, with both a simple single-resolution pre-training recipe and a two resolution training recipe mirroring DEIT III.

LieRE vs RoPE-Mixed generalization across resolutions (Please see the figure at https://imgur.com/a/bnPmERw)

| Resolution (imsize) | 192           	| 224           	| 256           	| 320           	| 384           	| 448           	|
|---------------------|-------------------|-------------------|-------------------|-------------------|-------------------|-------------------|
| liere           	| 64.6 (64.1, 65.0) | 69.3 (68.9, 69.7) | 71.4 (71.0, 71.8) | 73.2 (72.8, 73.6) | 73.1 (72.7, 73.5) | 71.7 (71.3, 72.1) |
| liere_finetune  	| 64.0 (63.5, 64.4) | 68.7 (68.3, 69.1) | 71.5 (71.1, 71.9) | 73.8 (73.4, 74.2) | 74.0 (73.6, 74.3) | 73.3 (72.9, 73.6) |
| naver           	| 63.7 (63.3, 64.2) | 68.8 (68.4, 69.2) | 71.1 (70.7, 71.5) | 72.5 (72.1, 72.9) | 72.1 (71.7, 72.5) | 69.7 (69.3, 70.1) |
| naver_finetune  	| 63.1 (62.7, 63.5) | 68.2 (67.8, 68.6) | 71.2 (70.8, 71.6) | 73.3 (72.9, 73.6) | 73.3 (72.9, 73.7) | 71.7 (71.3, 72.1) |


**Clarity and stylistic changes, particularly of the methods section (section 3):** We have rewritten the methods section to provide a clearer and more detailed explanation of LieRE, making it easier for readers to follow. We have also fixed numerous more minor issues throughout the paper.

We also identified a memory-related issue involving the layout of one of our parameter tensors after a reshape operation, which impacted certain experimental results, in particular learning across heads and layers for RoPE-Mixed and LieRE. We have since resolved this issue and include the top line results before and after the fix below. X’s indicate trainers that are in progress and we will update in the comments and the paper once the last trainer as finished. Additionally, we have thoroughly reviewed our codebase.  We also include the performance of LieRE with the constraint of using rotation matrices that are block diagonal with block size 64 and 8 instead shared across heads and layers based on the performance (Table 2). Due to compute restrains with changed the 3D depth to 8 (default in the Video MAE paper) instead of 32.

| Method | CIFAR-100 | ImageNet-1k | UCF101 |
|------------------------|--------------------------|-------------------------|------------------------|
| Abs. Pos. E.$^{*,**}$ | 63.9 (62.9-65.8) | 66.1 (65.7-66.5) | 40.9 (40.5-41.3) |
 | VisionLlama RoPE | 65.5 (64.6-66.5) | 65.4 (65.0-65.8) | 45.0 (44.6-45.4) |
| RoPE-Mixed | 68.8 (67.9-69.7) | 68.8 (68.4-69.2) | 46.3 (45.9-46.7) |
| LieRE$_{8}$ | **70.3 (69.4-71.2)** | **69.6 (69.2-70.0)** | **47.0 (46.6-47.4)** |
 | LieRE$_{64}$ | 70.0 (69.1-70.9) | 69.3 (68.9-69.7) | 44.7 (44.3-45.1) |

| Method            	| CIFAR-100       	| ImageNet-1k    	| UCF101           	|
|-----------------------|---------------------|--------------------|----------------------|
| Abs. Pos. E.*.**  	| 63.9 (62.9-65.8)   | 66.1 (65.7-66.5)   | 42.7 (42.3-43.1) 	|
| VisionLlama RoPE  	| 65.5 (64.6-66.5)   | 65.4 (65.0-65.8)   | 50.5 (50.1-50.9) 	|
| RoPE-Mixed        	| 63.7 (61.8-65.6)   | 65.9 (65.5-66.3)   | 45.7 (45.2-46.0) 	|
| LieRE-Commute_shared     	| 66.9 (66.0-67.9)   | 68.4 (67.9-68.9)   | 52.1 (51.7-52.5) 	|
| **LieRE_64_shared**         	| **69.9 (68.9-70.8)** | **69.5 (69.1-69.9)** | **53.6 (53.2-54.0)** |

We will upload an updated version of the paper once the final trainers complete so that all the differences are clearly visible in pdfdiff.

We believe these revisions not only address the reviewers' concerns but also strengthen the overall contribution of our work. We are excited about LieRE’s potential and its ability to advance positional encoding techniques for high-dimensional data and look forward to sharing with the field. Thank you again for your constructive feedback, and we look forward to any further comments or suggestions.

---

> ### Author Response · Authors · 2024-12-01
>
> We wanted to thank the reviewers for your constructive feedback. We are wondering if our newest comments address your concerns regarding:
>
> - **Writing style and clarity of the paper**: We improved the writing of the paper. In particular, we improved on the method section describing the LieRE.
> - **Resolutions Generalization**: We have included experiments to assess the method's generalization across resolutions compared to RoPE-Mixed. LieRE significantly outperforms RoPE-Mixed using both a straightforward single-resolution pre-training approach and a two-resolution training strategy mirroring the DEIT III training recipe.
>
> The discussion period is coming to an end tomorrow, and we hope the comments and new manuscript address the comments. If there are any remaining gaps, would it be possible for the reviewers to share them so we have a chance to address them?
>
> Thanks in advance!

---

### Meta-Review · Area_Chair_WRKk · 2024-12-24

**Metareview:**

Lie group Relative position Encodings (LieRE) is an alternative to or in a sense extension of Rotary Position Embeddings (RoPE). While RoPE learns a structured rotation matrices for keys and queries, LieRE learns general rotation matrices, and in effect has more degrees of freedom for the learned transformation. These degrees of freedom allow for the encoding of absolute and relative position information. Experiments on 2D image and 3D video classification show improved accuracy, training time, and data efficiency on the standard datasets of CIFAR-10/100, ImageNet and common but outdated dataset of UCF101. The empirical improvements and applicability to different dimensionalities are the key strengths, while lack of clarity and insufficient analysis are the key weaknesses. This work is missing a fully consistent analysis of results vs. relative encodings like RoPE, as some results are reported against absolute position baselines instead, while both are relevant to LieRE as an extension of both. This work is also missing sufficient clarity to be understood and informative to its audience, as evidenced by the two confirmed ratings for rejection and comments about level of detail and writing. The meta-reviewer highlights that the revision has made progress on this front, but at the same time the scale of its edits merits another round of review. The meta-reviewer sides with rejection on the balance, due to the issues of clarity and comparisons, although there is definite empirical promise for the proposed method. The authors are encouraged to further incorporate the reviewer feedback to resubmit a clearer and more thoroughly-analyzed edition of this work.

Note: the authors are advised to use parenthetical citations (`\citep`). This had no bearing on the decision, but is offered as miscellaneous typesetting advice.

**Additional Comments On Reviewer Discussion:**

The reviewers diverge in their evaluation: two vote for rejection (CJWD: 3, BRzg: 3) while one champions the paper (EcnH: 10). The authors provide a response to each review and a general summary. After the author-reviewer discussion, reviewers engaged in AC-reviewer discussion and updated their ratings. EcnH championed the paper and raised their rating from 8 to 10 due the practical impact and theoretical justification of the proposed method. In contrast CJWD lowered their score from 5 to 3 due to lack of details for clarity and lack of analysis for understanding of the results even if there is benchmark improvement. BRzg confirmed their rating of 3 due to remaining issues with clarity in the description of the method and a concern about the number of additional parameters needed by the proposed LieRE explaining away performance differences.

---

### Decision · Program_Chairs · 2025-01-22

Reject